# Stochastic Electrodynamics: Lessons from Regularizing the Harmonic Oscillator

**Theodorus Maria Nieuwenhuizen** 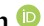

Institute for Theoretical Physics, 1098 XH Amsterdam, The Netherlands; t.m.nieuwenhuizen@uva.nl

**Abstract:** In this paper, the harmonic oscillator problem in Stochastic Electrodynamics is revisited. Using the exact shape of the Lorentz damping term prevents run-away effects. After introducing a cut-off in the stochastic power spectrum and regularizing the stochastic force, all relevant integrals are dominated by resonance effects only and results are derived that stem from those in the quantum ground state. For an orbit with specific position and momentum at an initial time, the average energy and the average rate of energy change are evaluated, which stem with each other. Resonance effects are highlighted along the way. An outlook on the hydrogen ground state problem is provided.

**Keywords:** stochastic electrodynamics; harmonic oscillator; harmonic oscillator ground state

---

## 1. Introduction

Quantum mechanics (QM) is a statistical theory, which does no less and no more than provide the Born probabilities for the outcomes of experiments. Though many interpretations have been put forward and various ontologies have been sought within the quantum theory, a deep analysis of the dynamics of a quantum measurement in a rich enough, but solvable, model has strengthened the case that QM provides no more than statistics. It is even capable of being connected to individual measurements, thus addressing the celebrated "measurement problem" [1,2]. However, in this philosophy, QM does not provide a framework to describe individual systems in detail.

In practice, in each single experiment, something occurs inside the apparatus which triggers the specific outcome for its pointer. It is our thesis that we do not have a theory for such events. The theory can not be QM, which provides only the statistics of outcomes; it is not classical electrodynamics, nor any other known theory. Nature produces individual events "every day, the whole day", and we are still lacking a theory for these. The challenge to find this theory is as large as it was to find special relativity and quantum mechanics in the early 1900s.

The theory of Stochastic Electrodynamics (SED) is a bold and attractive attempt to describe nature at a deeper level than by quantum theory, aiming to capture individual events. As such, it is an attempt to provide a sub-quantum mechanics. SED is a theory for classical particles embedded in a classical electromagnetic vacuum, with the intensity of its modes being compatible with the quantum zero-point spectrum. In particular, each vacuum mode is a plane wave with energy $\frac{1}{2}\hbar\omega$, where Planck's constant sets the energy scale of the fluctuation spectrum and appears as a new constant of nature. The SED theory has been well-developed; the reader is referred to the books [3,4]. Before addressing deeper questions, one, first, has to demonstrate that SED explains various properties of QM at the statistical level. This has been shown, on general grounds, within a certain flow of arguments [3,4]. Hence, specific cases are required to support the derivation and for gaining deeper insights. Progress has been made for harmonic oscillators [5–11], where a proper ground state emerges. Several phenomena related to oscillators have been explained as well, such as van der Waals forces, the Casimir effect, and the Unruh effect. Additionally, the leading logarithm of the Lamb shift between the hydrogen $1s$ and $2p$ states, has emerged in the harmonic oscillator and the hydrogen problems; however, the finite

part is not in agreement for the harmonic oscillator problem, and it has not been worked out for the hydrogen problem.

In a recent publication, we considered the hydrogen ground state in SED [12]. To lay the basis for a possible reformulation of that problem, we revisit, here, the harmonic oscillator problem, paying special attention to the prevention of peculiarities related to damping and ultraviolet divergences. We outline the problem in Section 2 and consider the average dynamical properties in Section 3. In Section 4, we consider the average progression from a general initial condition. We conclude with a summary.

## 2. The Basis of SED

The harmonic oscillator problem for an electron in SED is set by the stochastic differential equation

$$m\ddot{\mathbf{r}} = -m\omega^2\mathbf{r} - e\mathbf{E} + m\tau\dddot{\mathbf{r}}, \qquad \tau = \frac{2e^2}{3mc^2} = \frac{2\alpha}{3}\frac{\hbar}{mc^2} = 6.266\ 10^{-24}\ \text{s}, \tag{1}$$

where (in Gaussian units) $\alpha \approx 1/137$ is the fine-structure constant. We study this problem on both space and time scales, inspired by the Kepler problem for the hydrogen atom [12]. Hence, we consider a typical frequency $\omega_1$ and typical deviation $r_1 = \sqrt{\hbar/m\omega_1}$ such that $m\omega_1^2 r_1^2 = \hbar\omega_1$. This leads us to define the adimensional ("ad") time, distance, and adimensional charge $\beta$,

$$t_{\text{ad}} = \omega_1 t, \qquad \mathbf{r}(t) = r_1\mathbf{r}_{\text{ad}}(t_{\text{ad}}), \qquad \beta = \sqrt{\frac{2\alpha}{3}\frac{\hbar\omega_1}{mc^2}} > 0. \tag{2}$$

Noting that the dimension of $\mathbf{E}$ satisfies $\mathbf{E}(t) = \omega_1^2\sqrt{2\hbar/3c^3}\,\mathbf{E}_{\text{ad}}(t_{\text{ad}})$, we arrive at the field term

$$\frac{e\mathbf{E}(t)}{m\omega_1^2 r_1} = \beta\mathbf{E}_{\text{ad}}(t_{\text{ad}}). \tag{3}$$

Defining $\omega_0 = \omega/\omega_1$ and dropping the index "ad", we derive the adimensional equation of motion

$$\ddot{\mathbf{r}} = -\omega_0^2\mathbf{r} - \beta\mathbf{E} + \beta^2\dddot{\mathbf{r}}. \tag{4}$$

Though $\beta$ and $\omega_0$ are dimensionless, the combination $\omega\tau_c \to \omega_0\beta^2$ makes $\beta^2$ appear to have the dimension of time.

In the hydrogen problem, with a nucleus of charge $Ze$ one would have $\omega_0^2\mathbf{r} \to \mathbf{r}/r^3$. Furthermore, $\hbar\omega_1 = Ze^2/r_1 = Z^2\alpha^2 mc^2$ implies that $\omega_1 = Z^2\alpha^2 mc^2/\hbar$ and $r_1 = \hbar/Z\alpha mc$, and

$$\beta = \sqrt{\frac{2}{3}}\alpha^{3/2}Z \approx \frac{Z}{1965}. \tag{5}$$

We consider a time window, during which the orbit remains, for many revolutions, in the plane of the temporary orbit; which is possible due to the small value of $\beta$.

*Statistics of the Stochastic Electric Field*

The stochastic electric field $\mathbf{E}$ contains polarization indices. It is handy not to be bothered by this technical complication. In our present units, where $\mathbf{E}_{\text{ad}}(t_{\text{ad}}) = \sqrt{(3/2)(c^3/\hbar)}\,\mathbf{E}(t)/\omega_1^2$, we will keep the explicit factor $\sqrt{3/2}$ after $\mathbf{E}(t)$ is expressed in $t_{\text{ad}}$. Hence, in adimensional units, its spectrum can equally well be presented by the stochastic Gaussian field with the properties [13]

$$\mathbf{E}(t) = \int_{-\infty}^{\infty}\sqrt{D\omega}\,\mathbf{E}_\omega e^{-i\omega t}, \qquad D\omega = \frac{d\omega}{2\pi}|\omega|^3 e^{-|\omega|\tau_c}, \qquad \mathbf{E}_{-\omega} = \mathbf{E}_\omega^*, \qquad \langle\mathbf{E}_\omega\mathbf{E}_{\omega'}\rangle = \delta(\omega + \omega'). \tag{6}$$

Its correlator correctly represents the statistics of the EM field:

$$\mathbf{C}_{EE}(t-s) = \langle \mathbf{E}(t)\mathbf{E}(s) \rangle = \mathbf{1} \times C_{EE}(t-s),$$

$$C_{EE}(t-s) = \Re \int_{-\infty}^{\infty} \frac{d\omega}{2\pi} |\omega|^3 e^{-i\omega(t-s)-|\omega|\tau_c} = \frac{3}{2} \times \frac{4}{\pi} \Re \frac{1}{(t-s+i\tau_c)^4}. \tag{7}$$

Notice the factor of $(\sqrt{3/2})^2$. This notation and outcome can be verified from the frequency-discretization derived in [12]. It is expected that the cutoff $\tau_c$ can be taken to zero at the end. In the hydrogen problem, the small value of $\tau_c = \alpha^2 Z^2$ corresponds, in physical units, to the short Compton time $\hbar/mc^2$.

With $\mathbf{E} = -\partial_t \mathbf{A}$, the correlation function of the vector potential $\mathbf{A}$ follows from $C_{EE} = \partial_t \partial_s C_{AA}$, as

$$C_{AA}(t-s) = \frac{1}{\pi} \Re \frac{-1}{(t-s+i\tau_c)^2}. \tag{8}$$

At the end of Section 4, we will consider the primitive $\mathbf{C}$ of $\mathbf{A}$, and its correlation function

$$\mathbf{A} = \dot{\mathbf{C}}, \quad \mathbf{E} = -\ddot{\mathbf{C}}, \quad \langle \mathbf{C}(t)\mathbf{C}(s) \rangle = \mathbf{1} \times C_{CC}(t-s),$$

$$C_{CC}(t-s) = \Re \int_{-\infty}^{\infty} \frac{d\omega}{2\pi(|\omega|+\omega_{co})} e^{-i\omega(t-s)-|\omega|\tau_c} = -\frac{1}{\pi} \Re \log[\omega_{co}(t-s+i\tau_c)] - \frac{\gamma_E}{\pi}, \tag{9}$$

where $\omega_{co} \sim \alpha^3 \log 1/\alpha$ is an adimensional low-frequency cut-off, anticipated below (51), and $\gamma_E$ is Euler's constant.

### 3. The Harmonic Oscillator in SED and Its Steady State

The harmonic oscillator problem in SED has been studied by many leaders in the field; see, for example, [5–11], it is also discussed, at length, in the book [3]. This is typically done by taking, in frequency integrals, the contributions from resonances and not bothering much about high- or low-frequency peculiarities. It is our purpose to clarify where regularizations are needed and which form they should have, in order to derive these physically relevant results in a proper fashion.

Let the stochastic field and the particle position have the frequency representation

$$\mathbf{E}(t) = \int_{-\infty}^{\infty} \sqrt{D\omega}\, \mathbf{E}_\omega e^{-i\omega t}, \quad \mathbf{r}(t) = \int_{-\infty}^{\infty} \sqrt{D\omega}\, \mathbf{r}_\omega e^{-i\omega t}. \tag{10}$$

The equation of motion in time and frequency read, respectively,

$$\ddot{\mathbf{r}} = -\omega_0^2 \mathbf{r} - \beta \mathbf{E} + \mathbf{D}, \quad -\omega^2 \mathbf{r}_\omega = -\omega_0^2 \mathbf{r}_\omega - \beta \mathbf{E}_\omega + \mathbf{D}_\omega. \tag{11}$$

The damping term, if taken from Equation (4), would be $\mathbf{D}_\omega \approx i\beta^2 \omega^3 \mathbf{r}_\omega$. Fortunately, it has been derived, from first principles, in Equation (3.110) of [3]. In our notation, this exact result reads

$$\mathbf{D}(t) = \int_{-\infty}^{t} ds\, \dot{\mathbf{r}}(s)\overline{D}(t-s) = \int_{0}^{\infty} du\, \dot{\mathbf{r}}(t-u)\overline{D}(u), \quad \overline{D}(u) = -\frac{\beta^2}{\pi} \int_{-\infty}^{\infty} d\omega'\, \omega'^2 e^{-i\omega'u-|\omega'|\tau_c}, \tag{12}$$

where we assumed a similar exponential cut-off $\exp(-|\omega'|\tau_c)$ as in the stochastic spectrum. This structure is a truncated convolution, so that it still leads to a product in Fourier space. With $(\dot{\mathbf{r}})_\omega = -i\omega \mathbf{r}_\omega$, one gets

$$\mathbf{D}(t) = \int_{-\infty}^{\infty} \sqrt{D\omega}\, \mathbf{D}_\omega e^{-i\omega t}, \quad \mathbf{D}_\omega = \omega^2 D_\omega \mathbf{r}_\omega, \tag{13}$$

with the function $D_\omega$, not to be confused with $\mathbf{D}_\omega$ or the integration measure $D\omega$ of Equation (6), given by

$$
\begin{aligned}
D_\omega &= -\frac{i\omega}{\omega^2} \int_0^\infty du\, e^{i(\omega+i0)u} \overline{D}(u) = \frac{\beta^2}{\pi\omega} \int_{-\infty}^\infty d\omega'\, \frac{\omega'^2 e^{-|\omega'|\tau_c}}{\omega'-\omega-i0} = \frac{2\beta^2}{\pi} \int_0^\infty d\omega'\, \frac{\omega'^2 e^{-\omega'\tau_c}}{\omega'^2-(\omega+i0)^2} \\
&= \frac{2\beta^2}{\pi\tau_c} \int_0^\infty dx\, \frac{x^2 e^{-x}}{x^2-(\omega\tau_c+i0)^2} = i\beta^2\omega e^{-|\omega|\tau_c} + \frac{2\beta^2}{\pi\tau_c}\, \mathrm{PV} \int_0^\infty dx\, \frac{x^2 e^{-x}}{x^2-\omega^2\tau_c^2},
\end{aligned}
\tag{14}
$$

where PV denotes the principal value. Transforming to the time domain, this yields

$$
D(u) \equiv \int_{-\infty}^\infty \frac{d\omega}{2\pi} e^{-i\omega u} i\omega D_\omega = \theta(u)\overline{D}(u),
\tag{15}
$$

where $\theta$ is the Heaviside step function. Indeed, for $u < 0$, the contour can be closed in the upper half of the complex $\omega$ plane, where $i\omega D_\omega$ is analytic, as is evident from the middle expression in the first line of (14). Hence, $D(u) = 0$ for $u < 0$. Likewise, for $u > 0$, the contour can be closed in the lower half-plane, yielding $D(u) = \overline{D}(u)$. This explains the causality relation $D(t-s) = 0$ for $s > t$. After replacing $\overline{D} \to D$ in Equation (12) for $\mathbf{D}(t)$, we may extend the $s$ and $u$ integrals from $-\infty$ to $\infty$.

For small $\omega\tau_c$, the last expression in (14) yields

$$
D_\omega \approx \delta_m + i\beta^2\omega, \qquad \delta_m = \frac{2\beta^2}{\pi\tau_c}.
\tag{16}
$$

The $\delta_m$ term corresponds to a mass renormalization, due to the presence of the electromagnetic field modes. In the units of the hydrogen problem, $\delta m_e/m_e = \delta_m = 4\alpha/3\pi = 0.0031$, independent of $Z$ (as it should). The term $i\beta^2\omega$ corresponds to the Lorentz damping term $\mathbf{D} \approx \beta^2\dddot{\mathbf{r}}$ in (11). This approximation is known to have run-away solutions $\mathbf{r} \sim \exp t/\beta^2$, artifacts that are absent in our exact treatment. For large $|\omega|\tau_c$, one finds that the mass renormalization drops out and, instead, we have that $D_\omega \to -(4\beta^2/\pi\tau_c)/(\omega\tau_c)^2$ becomes negligible.

The solution of the inhomogeneous equation of motion (11) now reads

$$
\mathbf{r}_\omega = \beta G_\omega \mathbf{E}_\omega, \quad \mathbf{D}_\omega = \beta\omega^2 G_\omega D_\omega \mathbf{E}_\omega, \quad G_\omega = \frac{1}{\omega^2-\omega_0^2+\omega^2 D_\omega}.
\tag{17}
$$

The poles of $G_\omega$ determine the complex eigenfrequencies. They follow from

$$
\omega^2(1+D_\omega) = \omega_0^2.
\tag{18}
$$

Contrary to the approximation (16), which formally allows $D_\omega \sim -1$ at a large imaginary $\omega = i|\omega|$ with $|\omega| \sim (1+\delta_m)/\beta^2 + \beta^2\omega_0^2$, the exact function $D_\omega$ is small for all real and complex $\omega$, so there appear no such spurious eigenvalues and run-away solutions $\sim \exp(t/\beta^2)$ in the exact treatment. As expected on physical grounds, there are only solutions near $\pm\omega_0$. For small $\beta$ and $\delta_m$, they read

$$
\omega_\pm = \pm\overline{\omega}_0 - i\gamma, \qquad \overline{\omega}_0 = (1-\tfrac{1}{2}\delta_m)\omega_0, \qquad \gamma \approx \tfrac{1}{2}\beta^2\omega_0^2, \qquad \delta_m \approx \frac{2\beta^2}{\pi\tau_c},
\tag{19}
$$

with the approximations giving the leading terms in $\beta^2$.

*The Steady State*

We now get, from (6) and (17), the steady value

$$
\langle \mathbf{r}^2(t) \rangle = 3\beta^2 \int D\omega\, |G_\omega|^2 = \beta^2 \int_{-\infty}^\infty \frac{d\omega}{2\pi}\, \frac{|\omega^3| e^{-|\omega|\tau_c}}{(\omega^2-\omega_0^2)^2+(\beta^2\omega^3)^2},
\tag{20}
$$

where the first expression can be verified from the frequency-discretization of [13]. The integral is finite for $\tau_c \to 0$ and dominated by the narrow resonance region around $\omega_0$, with the result

$$\langle \mathbf{r}^2(t) \rangle = \frac{3}{2\omega_0}. \tag{21}$$

Likewise,

$$\langle \dot{\mathbf{r}}^2(t) \rangle = 3\beta^2 \int D\omega \, \omega^2 |G_\omega|^2 = 3\beta^2 \int_{-\infty}^{\infty} \frac{d\omega}{2\pi} \frac{|\omega^5| e^{-|\omega|\tau_c}}{|\omega^2 - \omega_0^2 + \omega^2 D_\omega|^2}. \tag{22}$$

There is a similar resonance around $\omega \approx \omega_0$, yielding $\langle \dot{\mathbf{r}}^2(t) \rangle = \frac{3}{2}\omega_0$ and $\mathcal{E}_0 = \frac{1}{2}\langle \dot{\mathbf{r}}^2(t) \rangle + \frac{1}{2}\omega_0^2 \langle \mathbf{r}^2(t) \rangle = \frac{3}{2}\omega_0$, which is the ground state energy of the $3d$ quantum oscillator. However, the large $\omega$ limit is only suppressed by the exponential. To evaluate its contribution to the leading order, $\omega_0^2$ can be set to zero, while $D_\omega \sim -(\beta^2/\tau_c)/(\omega \tau_c)^2$ can be neglected. The remaining integral in (26) is trivial, and brings

$$\langle \dot{\mathbf{r}}^2(t) \rangle = \frac{3}{2}\omega_0 + \frac{3\beta^2}{\pi\tau_c^2} = \frac{3}{2}\omega_0 + 3\beta^2 C_{AA}(0). \tag{23}$$

For $\omega_0 = 0$, this result follows immediately from the free-particle solution $(\dot{\mathbf{r}})_\omega = \beta \mathbf{A}_\omega/(1 + D_\omega)$, leading to $\dot{\mathbf{r}}(t) \approx \beta \mathbf{A}(t)$ and $\langle \dot{\mathbf{r}}^2(t) \rangle \approx 3\beta^2 C_{AA}(0)$. The large term $\beta^2 C_{AA}(0) \sim 1/\alpha Z^2$ comes from large frequencies $\sim 1/\tau_c$, which are cut off—but not enough—by the factor $\exp(-|\omega|\tau_c)$.

To further suppress the large $\omega$ contributions, we propose to subtract the free propagator $G_\omega^0$,

$$G_\omega^0 = \frac{1}{\omega^2 + \omega^2 D}, \qquad \overline{G}_\omega = G_\omega - G_\omega^0 \equiv \frac{1}{\omega^2 - \omega_0^2 + \omega^2 D} - \frac{1}{\omega^2 + \omega^2 D} = G_\omega \times \omega_0^2 G_\omega^0. \tag{24}$$

The expression $\mathbf{r}_\omega = \overline{G}_\omega \mathbf{E}_\omega$ can be written as $\mathbf{r}_\omega = G_\omega \overline{\mathbf{E}}_\omega$, with the renormalized stochastic field

$$\mathbf{E}_\omega \to \overline{\mathbf{E}}_\omega = G_\omega^{-1}(G_\omega - G_\omega^0)\mathbf{E}_\omega = (G_\omega^{0\,-1} - G_\omega^{-1})G_\omega^0 \mathbf{E}_\omega = \omega_0^2 G_\omega^0 \mathbf{E}_\omega = \frac{\omega_0^2}{\omega^2(1 + D_\omega)}\mathbf{E}_\omega. \tag{25}$$

Now, the extra factor $(\omega_0/\omega)^4|1 + D_\omega|^2$ assures enough suppression of the large $\omega$ contributions, so that the resonance at $\omega_0$ is dominant. This yields to the leading order

$$\langle \dot{\mathbf{r}}^2(t) \rangle = 3\beta^2 \int D\omega \, \omega^2 |\overline{G}_\omega|^2 = \frac{3}{2}\omega_0. \tag{26}$$

In the new expression for the position fluctuations,

$$\langle \mathbf{r}^2(t) \rangle = 3\beta^2 \int D\omega \, |\overline{G}_\omega|^2 = \frac{3}{2\omega_0} + \beta^2 \phi, \tag{27}$$

the correction $\phi$ inherits the logarithmic divergency at $\omega = 0$ from the free particle. This, likely, is addressed by accounting for soft photon emission, leading to a small $\beta^2 \log 1/\beta \sim \alpha^3 \log 1/\alpha$ "Lamb" correction.

The average energy now agrees, to leading order, with that of the ground state of the QM oscillator,

$$\mathcal{E} = \frac{3}{2}\omega_0. \tag{28}$$

## 4. Average Progression of a Specific Orbit

Following specific orbits in time reveals the structure of the dynamics. Due to the stochastic force, this can only be done numerically. The idea to look at the average progression of a collection of orbits, starting at some initial time at a general initial position and speed, was put forward, by de la Peña [14]

and Puthoff [15], for the hydrogen problem. Our numerical results [13,16] motivated us to revisit this average progression of a set of orbits [12]. To understand better what happens in this approach, we derive it, here, for the harmonic oscillator.

Suppose that the orbit has reached, at time $t = 0$, a position $\mathbf{r}_0$ and speed $\mathbf{v}_0$, and has energy

$$\mathcal{E}_0 = \frac{1}{2}\mathbf{v}_0^2 + \frac{1}{2}\omega_0^2\mathbf{r}_0^2. \tag{29}$$

The situation is described by adding a proper homogeneous solution to the previous inhomogeneous one. The solution with $\mathbf{r}(0) = \mathbf{r}_0$ and $\dot{\mathbf{r}}(0) = \mathbf{v}_0$ reads, for $t > 0$,

$$\mathbf{r}(t) = \mathbf{r}_c(t) + \beta \int \sqrt{\mathrm{D}\omega}\, \overline{G}_\omega \mathbf{E}_\omega e^{-i\omega t} r_\omega(t), \qquad \dot{\mathbf{r}}(t) = \dot{\mathbf{r}}_c(t) + \beta \int \sqrt{\mathrm{D}\omega}\, G_\omega \overline{\mathbf{E}}_\omega e^{-i\omega t} v_\omega(t),$$

$$\mathbf{r}_c(t) = e^{-\gamma t}(\mathbf{r}_0 \cos\overline{\omega}_0 t + \frac{\mathbf{v}_0 + \gamma\mathbf{r}_0}{\overline{\omega}_0} \sin\overline{\omega}_0 t), \quad \dot{\mathbf{r}}_c(t) = e^{-\gamma t}(\mathbf{v}_0 \cos\overline{\omega}_0 t - \frac{(\overline{\omega}_0^2 + \gamma^2)\mathbf{r}_0 + \gamma\mathbf{v}_0}{\overline{\omega}_0}\sin\overline{\omega}_0 t),$$

$$r_\omega(t) = 1 - e^{-\gamma t + i\omega t}(\cos\overline{\omega}_0 t + \frac{\gamma - i\omega}{\overline{\omega}_0}\sin\overline{\omega}_0 t), \tag{30}$$

$$v_\omega(t) = -i\omega + e^{-\gamma t + i\omega t}(i\omega \cos\overline{\omega}_0 t + \frac{\overline{\omega}_0^2 + \gamma^2 - i\gamma\omega}{\overline{\omega}_0}\sin\overline{\omega}_0 t),$$

where we have already inserted $G_\omega \mathbf{E}_\omega \to \overline{G}_\omega \mathbf{E}_\omega = G_\omega \overline{\mathbf{E}}_\omega$. As required, it holds that $r_\omega(0) = v_\omega(0) = 0$. Now, the expectation value of $\mathbf{r}^2$ already involves an integral of the form $\int \mathrm{D}\omega\, \omega^2 |\overline{G}_\omega|^2$, which, as we saw in Equation (26), is neatly dominated by the resonances at $\pm\overline{\omega}_0$.

Now consider the "state" as the ensemble of orbits with these initial conditions, progressing under all possible realizations of the stochastic field, with the appropriate weights. The average energy of this state is calculated, as above, by

$$\mathcal{E} = \mathcal{E}_{\mathrm{cl}} + \int_{-\infty}^{\infty} \mathrm{D}\omega |\omega_0^2 G_\omega^0 G_\omega|^2 \left(\frac{1}{2}|v_\omega|^2 + \frac{1}{2}\omega_0^2|r_\omega|^2\right). \tag{31}$$

To leading order in $\beta$, the classical orbit shrinks due to the radiation,

$$\mathcal{E}_{\mathrm{cl}} = e^{-2\gamma t}\mathcal{E}_0, \qquad \mathcal{E}_0 = \frac{1}{2}\mathbf{v}_0^2 + \frac{1}{2}\omega_0^2\mathbf{r}_0^2. \tag{32}$$

The frequency integrals are dominated by the resonances at $\pm\overline{\omega}_0$. We need the rule

$$\int_0^\infty \frac{\mathrm{d}\omega\, |\omega|^3 e^{-|\omega|\tau_c}}{\pi} \frac{\omega_0^4}{\omega^4 |1 + D_\omega|^2} \frac{a(\omega) + b(\omega)\,e^{\pm i\omega t}}{|\omega^2 - \omega_0^2 + \omega^2 D_\omega|^2} \approx \frac{\overline{\omega}_0}{4\gamma}\left[a(\overline{\omega}_0) + b(\overline{\omega}_0 \pm i\gamma)\,e^{\pm i\overline{\omega}_0 t - \gamma t}\right], \tag{33}$$

for functions $a$ and $b$ which are smooth around $\overline{\omega}_0$. The first terms of $r_\omega$ and $v_\omega$ in Equation (30) yield, in the integral (31), the steady state result again, while the absolute-squares of their second terms are of the type $a$ in (33). The cross-terms, finally, have one factor $e^{-\gamma t}$, but pick up another one because they are of the $b$-type in (33). Putting this all together, the total energy reads, if we neglect the $\gamma$ and $\delta_m$ corrections (except in the exponent), as

$$\mathcal{E}(t) = \frac{3}{2}\omega_0(1 - e^{-2\gamma t}) + \mathcal{E}_0 e^{-2\gamma t}, \tag{34}$$

which shows equilibration towards the average energy (28). The rate of energy change,

$$\dot{\mathcal{E}}(t) = 2\gamma\left(\frac{3}{2}\omega_0 - \mathcal{E}_0\right)e^{-2\gamma t}, \tag{35}$$

exhibits that orbits with $\mathcal{E}_0 > \frac{3}{2}\omega_0$ have the tendency to lose energy, and orbits with $\mathcal{E}_0 > \frac{3}{2}\omega_0$ to gain energy, again demonstrating the stability of the ground state. The adimensional characteristic timescale is $1/(2\gamma) = 1/(2\beta^2\omega_0^2)$, which reads $1/(2\omega^2\tau)$ in the physical units of Equation (1).

### 4.1. Average Change of Energy of a Specific Orbit at Moderate Times

For the hydrogen problem, we have found numerical and analytical support for the thesis that SED does not produce a proper stationary state, but instead leads to self-ionization. For analyzing such cases, the best one can do is to evaluate the average rate of energy change of sets of specific orbits. Let us, therefore, derive this here, in detail, for the harmonic oscillator, with the goal of confirming Equation (35).

The change of energy is, by definition,

$$\dot{\mathcal{E}} = \langle(\ddot{\mathbf{r}} + \omega_0^2\mathbf{r})\cdot\dot{\mathbf{r}}\rangle = \dot{\mathcal{E}}_{\text{field}} - \dot{\mathcal{E}}_{\text{rad}}, \qquad \dot{\mathcal{E}}_{\text{field}} = -\beta\langle\mathbf{E}\cdot\dot{\mathbf{r}}\rangle, \quad \dot{\mathcal{E}}_{\text{rad}} = -\langle\mathbf{D}\cdot\dot{\mathbf{r}}\rangle. \tag{36}$$

In the term $\dot{\mathcal{E}}_{\text{field}}$, the integrand diverges before regularization as $\omega^3\exp(-|\omega|\tau_c)$ for large $|\omega|$. To regularize this, we need to insert two factors $\omega_0^2 G_\omega^0$, corresponding to evaluating $-\beta\langle\mathbf{E}\cdot\dot{\mathbf{r}}\rangle$ with, again, both its factors $\mathbf{E}_\omega$ replaced by $\overline{\mathbf{E}}_\omega$. For $\dot{\mathbf{r}}$, this was already done—see Equation (30).

### 4.2. The Energy Gain Term

Taking the contributions from $\omega < 0$ and $\omega > 0$ together, we evaluate

$$\overline{\dot{\mathcal{E}}_{\text{field}}} \;=\; \frac{6\gamma}{\omega_0^2}\int_0^\infty \frac{d\omega}{\pi}\omega^3 e^{-\omega\tau_c}|\omega_0^2 G_\omega^0|^2\,\Re[-G_\omega v_\omega(t)]. \tag{37}$$

The propagator $G$ has poles in the lower half plane, $G \approx 1/[(\omega + \omega_0 + i\gamma)(\omega - \omega_0 + i\gamma)]$. For $t > 0$, we can close the contour in the upper quarter plane, and integrate along the imaginary $\omega = iz$ axis

$$\overline{\dot{\mathcal{E}}_{\text{field}}} \;=\; \frac{6\gamma}{\omega_0^2}\int_0^\infty \frac{dz}{\pi}z^3\cos(z\tau_c)|\omega_0^2 G_{iz}^0|^2\,\Re[-G_{iz}v_{iz}(t)] \tag{38}$$

$$=\; 6\gamma\omega_0^2\int_0^\infty\frac{dz}{\pi}\frac{\cos z\tau_c}{z^2+\omega_0^2}[1 - e^{-\gamma t - zt}(\cos\overline{\omega}_0 t - \frac{\overline{\omega}_0}{z}\sin\overline{\omega}_0 t)], \tag{39}$$

where we skipped the $\gamma \sim \beta^2$ corrections in $v_{iz}$ and in $G_{iz} \approx -1/(z^2 + \omega_0^2)$. Without needing the cut-off term, which has become $\cos z\tau_c$ here, we can express this as

$$\overline{\dot{\mathcal{E}}_{\text{field}}} \;=\; 3\gamma\omega_0 - 6\gamma\omega_0 e^{-\gamma t}\int_0^\infty\frac{dy}{\pi}\frac{e^{-y\omega_0 t}}{y^2+1}(\cos\overline{\omega}_0 t - \frac{1}{y}\sin\overline{\omega}_0 t). \tag{40}$$

The $\sin\overline{\omega}_0 t$ term has a logarithmic singularity at $y = 0$. Averaging over one period, from $t$ to $t + P$ with period $P = 2\pi/\overline{\omega}_0$, this singularity drops out, yielding to leading order in $\gamma$

$$\overline{\dot{\mathcal{E}}_{\text{field}}} \;=\; 3\gamma\omega_0 - 6\gamma\omega_0 e^{-\gamma t}\int_0^\infty\frac{dy}{\pi}\frac{e^{-y\omega_0 t}}{y^2+1}\frac{1 - e^{-2\pi y}}{2\pi y}\frac{(y^2-1)\cos\omega_0 t - 2y\sin\omega_0 t}{y^2+1}. \tag{41}$$

The integral decays at $1/\omega_0 t$, so that

$$\overline{\dot{\mathcal{E}}_{\text{field}}} \;=\; 3\gamma\omega_0(1 + \frac{2}{\pi}\frac{e^{-\gamma t}}{\omega_0 t}\cos\omega_0 t) \to 3\gamma\omega_0 \tag{42}$$

decays, in an algebraic and damped oscillatory fashion, to its ensemble average.

### 4.3. The Energy Loss Term

For the energy loss by radiation, the contribution from the classical orbit with damping $\mathbf{D}_c = \beta^2 \dddot{\mathbf{r}}_c$ reads

$$\mathcal{E}_{\text{rad}}^{(\text{cl})} = -\mathbf{D}_c \cdot \dot{\mathbf{r}}_c = -\beta^2 \dddot{\mathbf{r}}_c \cdot \dot{\mathbf{r}}_c = -\beta^2 \frac{d}{dt}(\ddot{\mathbf{r}}_c \cdot \dot{\mathbf{r}}_c) + \beta^2 \ddot{\mathbf{r}}_c^2. \tag{43}$$

Averaged over one period, this brings, for $\beta \ll 1$ and $\gamma \ll \omega_0$, the radiation damping effect

$$\overline{\dot{\mathcal{E}}_{\text{rad}}^{(\text{cl})}} = -\overline{\mathbf{D}_c \cdot \dot{\mathbf{r}}_c} = \beta^2 \overline{\ddot{\mathbf{r}}_c^2} = \beta^2 \omega_0^4 \overline{\mathbf{r}_c^2} = \beta^2 \omega_0^2 e^{-2\gamma t}(\frac{1}{2}\mathbf{v}_0^2 + \frac{1}{2}\omega_0^2 \mathbf{r}_0^2) = 2\gamma e^{-2\gamma t}\mathcal{E}_0, \tag{44}$$

which depends on $\mathbf{r}_0$ and $\mathbf{v}_0$ only through $\mathcal{E}_0$. The fluctuation contribution to this takes the form

$$\dot{\mathcal{E}}_{\text{rad}}^{(\text{fl})} = 6\beta^2 \int_0^\infty D\omega |\omega_0^2 G_\omega^0 G_\omega|^2 \, \Re[-\omega^2 D_\omega r_\omega(t) v_\omega^*(t)]. \tag{45}$$

With $D_\omega = \beta^2(\frac{\pi}{2\tau_c} + i\omega) \sim \gamma$, this expression is formally of order $\gamma^2$; however, the resonances connected to contributions from many previous orbits, bring a factor, when summed, of the order $\omega_0/\gamma$. The integral is dominated by the imaginary part of $D_\omega$. Indeed, this integral yields

$$\dot{\mathcal{E}}_{\text{rad}}^{(\text{fl})} = 3\gamma\omega_0(1 - e^{-2\gamma t}). \tag{46}$$

### 4.4. The Energy Balance for a Specific Orbit

We see that the rate of energy gain from the stochastic field quickly goes to its steady value $3\gamma\omega_0$, while the fluctuation contribution to the energy radiation starts out as zero, and grows as $1 - e^{-2\gamma t}$.

Combining (42) with (44) and (46), the average rate of total energy change of the state starting at $t = 0$ with energy $\mathcal{E}_0$ is to leading order in the small parameter $\gamma \sim \beta^2$

$$\overline{\dot{\mathcal{E}}} = \dot{\mathcal{E}}_{\text{field}} - \dot{\mathcal{E}}_{\text{rad}} = [3\gamma\omega_0] - [2\gamma e^{-2\gamma t}\mathcal{E}_0 + 3\gamma\omega_0(1 - e^{-2\gamma t})] = 2\gamma(\frac{3}{2}\omega_0 - \mathcal{E}_0)\,e^{-2\gamma t}. \tag{47}$$

This is in accordance with (35), derived from the time-dependent energy.

### 4.5. Renormalized Force in the Temporal Description

In order to avoid high-frequency peculiarities, we had to renormalize the noise—see Equation (25). The renormalized equation of motion,

$$\mathbf{r}_\omega = \beta(G_\omega - G_\omega^0)\mathbf{E}_\omega, \tag{48}$$

implies $G_\omega^{-1}\mathbf{r}_\omega = \beta(G_\omega^{0\,-1} - G_\omega^{-1})G_\omega^0 \mathbf{E}_\omega$; that is to say

$$(\omega^2 - \omega_0^2 + \omega^2 D_\omega)\mathbf{r}_\omega = \beta\omega_0^2 G_\omega^0 \mathbf{E}_\omega = \frac{\beta\omega_0^2 \mathbf{E}_\omega}{(1 + \delta_m)\omega^2}(1 + \mathcal{O}(\beta^2)). \tag{49}$$

In the time domain, this reads (dropping the $\mathcal{O}(\beta^2)$ corrections)

$$\ddot{\mathbf{r}} = -\omega_0^2 \mathbf{r} - \beta\omega_0^2 \mathbf{C} + \mathbf{D}, \tag{50}$$

where the stochastic potential $\mathbf{C}(t)$ is defined by $\ddot{\mathbf{C}}(t) = -\mathbf{E}(t)/(1 + \delta_m)$. It has covariance

$$\langle \mathbf{C}(t)\mathbf{C}(s) \rangle = \mathbf{1} \times C_{CC}(t - s), \qquad C_{CC}(t - s) = \frac{1}{(1 + \delta_m)^2} \int_{-\infty}^\infty \frac{d\omega}{2\pi} \frac{e^{-|\omega|\tau_c}}{|\omega| + \omega_{co}} e^{i\omega(t-s)}. \tag{51}$$

Here, we tentatively cut off the new divergency at $\omega = 0$ by adding $\omega_{co}$ into the denominator. The equation of motion (50) follows from (11) by setting $\mathbf{r} \rightarrow \mathbf{r} + \beta^2 \mathbf{C}$ and neglecting the induced corrections to the small damping $\mathbf{D}$.

## 5. Summary: Lessons from the Harmonic Oscillator Analysis

Equation (50) can be viewed as the motion of the particle in the stochastic potential $V(\mathbf{r}, t) = V(\mathbf{r} + \beta \mathbf{C}(t)) = \frac{1}{2}\omega_0^2[\mathbf{r} + \beta \mathbf{C}(t)]^2$. For the hydrogen problem, this approach was considered in Section 2.3.1 of [13]; the formulation in which was used as a check for our numerics.

The important lessons we learned for this harmonic oscillator problem were:

(1)   The noise must be renormalized to suppress the dominance of high frequencies;
(2)   the damping has to be treated non-perturbatively to set the width of the resonance window;
(3)   the energy absorbed from the field quickly goes to its ensemble average; and
(4)   the energy radiation contains effects from both the classical orbit and the stochastic field.

Though the latter is formally of order $\beta^4$, it contributes to the leading $\beta^2$ behavior due to the resonances. It starts out at 0 and decays to its ensemble average over a few damping periods.

Notice, however, that if the regularization (24) (or 25)) is not adopted, the average kinetic energy can become very large—see Equation (23). Such excessive behavior arises from energy injection by high-frequency modes, a general aspect of SED, be it theoretical or numerical.

Noise renormalization is also important for the energy radiation rate. If one calculates the fluctuating part of $\beta^2 \langle \ddot{\mathbf{r}}(t) \rangle$ by using the equation of motion (11), one observes a large term $\beta^4 \langle \mathbf{E}(t)^2 \rangle = (6/\pi)\beta^4/\tau_c^4 \sim 1/\alpha^2 Z^4$, which does not disappear when one solves the same effect from the same equation in an equivalent way; however, when employing the renormalized equation of motion (50) it yields, at most (actually, not even), a logarithmic divergency, through $\beta^4 \langle \mathbf{C}^2(t) \rangle$.

We hope that these insights will improve the understanding of other properties of SED; in particular, for the hydrogen ground state. In that problem, it has been established that certain fluctuation modes are secular (i.e., growing linearly in time). Clearly, this leads, formally, to corrections which relatively quickly exceed the leading order effects. It appears that, at the linear level, these secular terms can be absorbed in the unperturbed orbit by taking its angle in the plane of the orbit at a slightly modified value, through $\phi \rightarrow \phi' = \phi + \delta\phi$. Working with this expression corresponds to taking the effect to all orders. The non-secular fluctuations are bounded, as in the harmonic oscillator problem, and will likely lose their correlation with $\phi'$ quite quickly. It is yet to be investigated whether the above lessons make the non-secular fluctuations well-behaved and, ideally, provide a regularization that makes the hydrogen ground state problem in SED physically sound.

**Funding:** This research received no external funding.

**Acknowledgments:** This work is inspired by the workshop on stochastic electrodynamics, SED2018 in Boston, 18–20 July 2018. It is a pleasure to thank the organizers Herman Batelaan, Ana–Maria Cetto, and Daniel Cole for the invitation and for creating a stimulating atmosphere.

**Conflicts of Interest:** The author declares no conflict of interest.

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
