# Peer review of "Stochastic Electrodynamics: Lessons from Regularizing the Harmonic Oscillator"

_atoms, doi:10.3390/atoms7020059_

Round 1

Reviewer 1 Report

The paper studies some aspects of the harmonic oscillator from the perspective of Stochastic Electrodynamics. Though this problem has been largely studied before, the author tackles it employing different and more precise mathematical techniques, focuses on important features of the SED approach —such as the resonance effects and the stability of the (ground) state, ultimately due to the presence of the stochastic field— and contributes with novel expressions related to the rate of energy change. 

The paper contains interesting results, yet it is somehow cryptic, mainly along the mathematical derivations and physical approximations, which in some cases may be difficult to follow (particularly for the broad, non-specialized audience). In this sense, additional arguing and references throughout the derivations and approximations made would help the reader to better grasp the author’s thinking. Furthermore, references to some of the (many) previous studies of the harmonic oscillator within SED should be included.

Also, there are some mathematical symbols that should be defined in the text (or defined in the appropriate place, somewhere near the corresponding expression). These are:

— \tau in Eqs. (2.3), (2.4)

— \tau_c in Eq. (2.4)

— \omega_c in Eq. (2.6). 

— PV (Principal value) in Eq. (3.5)

— D_c in Eq. (4.15)

I suggest the author to check that all symbols are properly defined along the manuscript.

Additional observations regarding the presentation are:

— It would be nice to clarify the meaning of the lines below Eq. (2.2), referring to the time window (as it is now it seems obscure).

—I find a problem with the units of \beta. In Eq. (2.2) it appears as a dimensionless quantity, yet in other equations (for example, in the expression for \gamma, Eq. (3.10)) it seems that [\beta^2]=[1/\omega].

— Which are the surprises the author refers to just before Eq. (2.6)? The phrase is unclear to me.

— Eq. (2.6) seems to be out of place there…its purpose and physical meaning are not clear.

Further, the author states that `SED is a classical theory having a classical electromagnetic vaccum, …’. This seems a rather confusing statement, because the zero-point stochastic field —the main ingredient of SED— is foreign to the domain of classical physics, so how can a theory that incorporates non-classical elements be itself a classical theory? A comment on this would be appreciated.

Regarding the expression for the rate of energy change showing the tendency towards the ground state, I think it would be nice to include some discussion on the time needed to reach the stable orbit, and its comparison with the mean lifetimes.

Finally, I find convenient to enrich the final summary and add some closing comments discussing in which specific direction the results presented here may be of relevance for a better understanding of the H atom problem within SED, and more generally, how the present analysis contributes (if so) to a better understanding of the sub-quantum behaviour.   

I think the above suggestions will serve to improve this manuscript --whose content is certainly interesting-- and recommend its publication in Atoms.

Author Response

I thank the referee very much for the many thoughtful advises. They have certainly chelped me at the relevant locations to improve the readability of the manuscript. In particular:

- I added extra introductory lines in various sections

- I added 7 references to previous investigations

- the listed typos are corrected and the symbols clarified where needed

- the lines below the old eq (2.2) are clarified.

- The "surprises" are eliminated. The role of the old eqn (2.6) is clarified.

- The statement about "SED'' as a "classical theory" is reformulated.

- the derivation of the dimensionless harmonic oscillator problem is carried out from its dimension-full version, the new (2.1)

- The timescale for loss of memory about the initial state is discussed.

- An informative outlook for the hydrogen problem is added at the end

Reviewer 2 Report

The paper could be published in present form, but I think it could be made more readable for people who do not work in this area if the author addresses the following small points:

Be explicit about the (Bohr) units used in writing Eq (2.1).

Define/discuss tau in (2.3).

Define/discuss tau_s in (3.3).

A typo: Replace "loose" by "lose" after (4.7).

Author Response

I thank the referee for the all-too-relevant remarks (s)he made.

- the derivation of the harmonic oscillator problem in Bohr units is carried out from its dimension-full version, the new (2.1)

- the tau and tau_s have all been clarified.

- the typo has been corrected.